# Rous Sarcoma Virus Genomic RNA Dimerization Capability In Vitro Is Not a Prerequisite for Viral Infectivity

**DOI:** 10.3390/v12050568

**Published:** 2020-05-22

**Authors:** Shuohui Liu, Rebecca Kaddis Maldonado, Tiffiny Rye-McCurdy, Christiana Binkley, Aissatou Bah, Eunice C. Chen, Breanna L. Rice, Leslie J. Parent, Karin Musier-Forsyth

**Affiliations:** 1Department of Chemistry and Biochemistry, Center for Retroviral Research, and Center for RNA Biology, The Ohio State University, Columbus, OH 43210, USA; liu.4982@osu.edu (S.L.); rye-mccurdy.1@osu.edu (T.R.-M.); cgbinkley@wisc.edu (C.B.); bah.44@buckeyemail.osu.edu (A.B.); 2Departments of Medicine, Division of Infectious Diseases and Epidemiology, Penn State College of Medicine, Hershey, PA 17033, USA; rjk297@psu.edu (R.K.M.); exc281@psu.edu (E.C.C.); blr190@psu.edu (B.L.R.); lparent@psu.edu (L.J.P.); 3Department of Microbiology & Immunology, Penn State College of Medicine, Hershey, PA 17033, USA

**Keywords:** Rous sarcoma virus (RSV), retroviruses, Gag polyprotein, packaging signal (Ψ), selective 2′-hydroxyl acylation analyzed by primer extension (SHAPE), RNA secondary structure, RNA structure-probing, dimerization initiation signal (DIS), RNA–protein interactions, UV crosslinking-coupled SHAPE (XL-SHAPE)

## Abstract

Retroviruses package their full-length, dimeric genomic RNA (gRNA) via specific interactions between the Gag polyprotein and a “Ψ” packaging signal located in the gRNA 5′-UTR. Rous sarcoma virus (RSV) gRNA has a contiguous, well-defined Ψ element, that directs the packaging of heterologous RNAs efficiently. The simplicity of RSV Ψ makes it an informative model to examine the mechanism of retroviral gRNA packaging, which is incompletely understood. Little is known about the structure of dimerization initiation sites or specific Gag interaction sites of RSV gRNA. Using selective 2′-hydroxyl acylation analyzed by primer extension (SHAPE), we probed the secondary structure of the entire RSV 5′-leader RNA for the first time. We identified a putative bipartite dimerization initiation signal (DIS), and mutation of both sites was required to significantly reduce dimerization in vitro. These mutations failed to reduce viral replication, suggesting that in vitro dimerization results do not strictly correlate with in vivo infectivity, possibly due to additional RNA interactions that maintain the dimers in cells. UV crosslinking-coupled SHAPE (XL-SHAPE) was next used to determine Gag-induced RNA conformational changes, revealing G218 as a critical Gag contact site. Overall, our results suggest that disruption of either of the DIS sequences does not reduce virus replication and reveal specific sites of Gag–RNA interactions.

## 1. Introduction

Rous sarcoma virus (RSV) is an alpharetrovirus that causes tissue tumors in chicken [1]. Like other retroviruses, in the early stage of the viral life cycle, RSV reverse transcribes its single-stranded genomic RNA (gRNA) into a double-stranded cDNA, which is then integrated into the host cell chromosome [2]. Subsequently, the integrated provirus hijacks the cellular machinery to perform transcription, nuclear export and translation [2]. The unspliced, full-length gRNA has two roles: one is to serve as the mRNA for the translation of Gag and Gag-pol polyproteins; another is to be packaged into the newly generated viral particles [2]. Similar to other retroviruses, the packaging of RSV gRNA is initiated by the interaction between the nucleocapsid (NC) domain of Gag and a packaging signal (Ψ), which is located in the 5′-untranslated region (UTR) of the gRNA [3,4]. Retroviral gRNA 5′-UTRs are highly structured and regulate numerous aspects of the viral life cycle, such as gRNA dimerization, packaging, reverse transcription and translation [5]. 

RSV gRNA has a contiguous, well-defined Ψ element (designated as mega-Ψ or MΨ, nucleotides (nt) 156–315 of the Prague C genome), which can direct the packaging of heterologous (i.e., non-native) RNAs with very high efficiency [6,7]. Subsequent studies identified a minimal sequence that is required for efficient packaging, micro-Ψ (µΨ, nt 156–237), which contains a stem region and three stem loops, SL-A, SL-B and SL-C [8,9]. In the case of human immunodeficiency virus type 1 (HIV-1), a minimal contiguous Ψ sequence has not been clearly identified. Although a minimal “core encapsidation signal” sequence has been proposed [10,11], other studies showed that *gag* coding sequences are additionally required for efficient packaging [12,13]. The simplicity of RSV Ψ makes it an informative model to examine the mechanisms of gRNA packaging and Gag–Ψ interactions. 

RSV Gag contains four major domains: matrix (MA), capsid (CA), NC and protease (PR). The MA domain binds to the membrane during viral assembly, and possesses weak nucleic acid binding activity [14,15,16]. The CA domain triggers Gag–Gag interactions and facilitates the formation of viral cores [17,18,19]. NC interacts with gRNA to initiate packaging and exhibits nucleic acid chaperone activity [3,16,19,20], and PR is involved in viral maturation [21]. RSV Gag also contains p2, p10 and spacer peptide (SP) domains, which function in budding [22,23], mature virion morphology and nuclear export [24], and the proper assembly of immature virions [25,26], respectively. RSV NC has two zinc finger motifs, which are indispensable for the specific interactions between RSV NC and Ψ [27,28,29]. The basic residues of NC also play an important role in Ψ RNA binding [30,31]. 

Numerous studies have been carried out to determine the key elements within RSV Ψ that are important for specific gRNA packaging. Cell-based packaging assays revealed that the single-stranded region between SL-A and SL-B is critical for gRNA packaging [8,9]. Residue A197 within this region was proposed to be involved in direct NC binding [32]. The loop region of SL-C was also shown to be essential for high-affinity NC binding [32]. A structure of an NC–µΨ complex solved by nuclear magnetic resonance (NMR) showed that A197 and G218 in the SL-C loop directly interact with RSV NC [33]. Mutations of these two interaction sites also partially or completely abolished viral infectivity [33]. Earlier cell-based packaging studies indicated that the structures, instead of sequences of the stems of SL-A and SL-B, are critical for Gag binding and gRNA packaging [8,9]. However, replacement of the SL-C tetraloop (UGCG to UUUG) did not affect RNA packaging significantly [8]. In vivo selection studies also showed that a randomized SL-C stem loop still allows the production of infectious virions [34]. Taken together with the viral infectivity assays, these studies suggested that A197 plays a primary role in Gag binding and gRNA packaging, while the SL-C tetraloop may only be of secondary importance [8,33]. The stem at the base of µΨ is highly conserved and important for genome packaging [8,35]. The NMR structure showed that this element is not directly involved in NC binding, but instead stabilizes the overall tertiary structure of µΨ [33]. 

A hallmark of retroviruses is that gRNAs are packaged as dimers (reviewed in [36]). Dimerization is initiated by a dimerization initiation signal (DIS) that contains a palindromic sequence, enabling intermolecular kissing-loop interactions [36]. Not surprisingly, electron microscopy studies of RSV gRNAs extracted from virions showed that they were present as dimers [37]. Previous in vitro dimerization studies in RSV characterized a dimer linkage structure (DLS), which was historically the name used to represent regions involved in gRNA dimerization within the *gag* coding region [37,38,39]. Studies focusing on avian leukosis virus (ALV), another alpharetrovirus that is closely related to RSV, suggested that a palindromic sequence in the L3 stem loop located downstream of SL-C is important for ALV gRNA dimerization [40,41]. However, an antisense oligonucleotide directed against the L3 loop of RSV RNA only weakly prevented dimerization [40]. Therefore, the exact identity of the RSV gRNA DIS element(s) remains to be determined. 

Despite the numerous studies investigating RSV gRNA recognition, packaging and dimerization, little is known about the secondary structure of RSV gRNA. Here we used selective 2′-hydroxyl acylation analyzed by primer extension (SHAPE) to solve the secondary structure of the 636-nt RSV 5′-leader RNA. The new structural information was used to predict a possible DIS candidate, and the role of these elements for gRNA dimerization and infectivity was tested. We also probed Gag–5′-leader RNA interactions using crosslinking-coupled SHAPE (XL-SHAPE) [42]. The putative Gag interaction sites were tested via mutational studies. Overall, our results provide new insights into RSV gRNA secondary structure, dimerization and Gag interactions with important implications for retroviral gRNA packaging.

## 2. Materials and Methods 

### 2.1. Preparation of Proteins and RNAs

RSV Gag∆PR protein was expressed and purified from *Escherichia coli*. The plasmid construct, pET28(-His).RSVGag∆PR, derived from the Prague C strain of RSV [43,44,45], contains MA through NC domains of the RSV Gag polyprotein (Figure 4A). The protein was purified as previously described [45] with the following alterations. After purification using phosphocellulose (PC) resin (from Sigma Aldrich, St. Louis, MO, USA), fractions containing Gag were combined and the concentration of monovalent ion was reduced to 100 mM. The protein was loaded onto a HiTrap Heparin HP affinity column (GE Healthcare, Chicago, IL, USA) and eluted in 0.2–0.8 M NaCl, 20 mM Tris-HCl, pH 7.4, 10 mM 2-mercaptoethanol (βME) and 1 µM ZnCl_2_. Peak fractions were collected and examined using sodium dodecyl sulfate–polyacrylamide gel electrophoresis (SDS-PAGE) and the purist fractions were combined and dialyzed into 10 mM 4-(2-hydroxyethyl)-1-piperazineethanesulfonic acid (HEPES), pH 7.5, 500 mM NaCl, 0.1 mM ZnCl_2_, 0.1 mM ethylenediaminetetraacetic acid (EDTA) and 1 mM βME. The protein concentration was determined by measuring the absorbance at 280 nm using a molar extinction coefficient of 63,348 M^−1^∙cm^−1^. A representative gel showing the purity of the protein is shown in Appendix A.

All RNA constructs were prepared by in vitro transcription using linearized plasmid templates and T7 RNA polymerase, as described [46]. The DNA sequences encoding the wild-type (WT) 636-nt RSV 5′-leader, MΨ-WT and RSV 167 RNAs were synthesized and cloned into pIDTSMART vector by Integrated DNA Technologies (IDT). Each of the RNAs was designed with two additional guanosine residues at the 5′-end to promote efficient in vitro transcription. RSV 5′-leader RNA was derived from nt 1–634 of RSV Prague C strain, and the template plasmid was linearized by XhoI. MΨ-WT and RSV 167 RNAs were derived from nt 156–315 and nt 1249–1409 of RSV Prague C strain, respectively. Additional sequences (AAGCU) were added to the 3′-end of these RNAs to facilitate the linearization of the template plasmid by HindIII. RSV 5′-leader RNA dimerization mutants (∆DIS-1, ∆DIS-2, ∆DIS-0, DIS-DM, DIS-TM, ∆L3, ∆SL-A and ∆L3+SL-A, Figure 2A) and MΨ RNA mutants (G218C, A197C, AAA, AACA, ∆L3 and ∆SL-B, Figure 5A) were prepared using site-directed ligase-independent mutagenesis (SLIM) [47]. The concentrations of all the RNAs were determined by measuring the absorbance at 260 nm and the following molar extinction coefficients: 5.9 × 10^6^ M^−1^∙cm^−1^ (RSV 5′-leader WT, ∆DIS-1, ∆DIS-2, ∆DIS-0, DIS-TM), 5.8 × 10^6^ M^−1^∙cm^−1^ (RSV 5′-leader DIS-DM, ∆SL-A), 5.7 × 10^6^ M^−1^∙cm^−1^ (RSV 5′-leader ∆L3), 5.6 × 10^6^ M^−1^∙cm^−1^ (RSV 5′-leader ∆L3+SL-A) and 1.5 × 10^6^ M^−1^∙cm^−1^ (RSV 167, MΨ-WT and MΨ mutants).

RSV 167, MΨ-WT and mutant RNAs were labeled with fluorescein-5-thiosemicarbazide (FTSC) at the 3′-end, as described [48]. The labeling efficiency was determined by measuring the absorbance at 495 nm, using a molar extinction coefficient of 8.5 × 10^4^ M^−1^∙cm^−1^. 

To create the DIS mutant proviruses used for infectivity studies, the 5′ LTR of RC.V8 was exchanged with that of pAT.V8 (WT or mutant DIS) using Gibson assembly [49]. For fragment 1, pRC.V8 was digested with *FseI* and *BsmI* to remove the 5′ LTR and part of the *gag* coding region from MA to p10. Fragment 2, encompassing the pAT.V8 and half of the MA coding region (either WT or mutant), was amplified using primers 5′-GCA TGC GCA TTC ATG GGC CAT TTT ACC ATT CACC-3′ and 5′-GAG CCG CCT TCA ATG CC-3′. Fragment 3 encompassing the pRC.V8 *gag* coding region extending from MA to p10 was amplified using 5′-GCT ATG ATA CTT GGG AAA TCG-3′ and 5′-CCT TGC CCA GTC AGT CAG-3′. Fragment 4 encompassing the pRC.V8 5′LTR U3 was amplified using 5′-GAT GGC GGA CGC GATG-3′ and 5′-GTG GTG AAT GGT AAA ATG GCG TTT ATT GTA TCG AGC TAG GC-3′.

To create the DIS deficient pRC.V8 Bal31 mCherry construct, we started with pGEM.RSVLTR.15-4, which contains a 77-bp deletion between nucleotides 219 and 296 generated by nuclease cleavage using Bal31, as previously described in [50]. The deletion was inserted into pRC.V8 using three fragments via Gibson assembly [49]. Fragment 1 generated the backbone by digesting pRC.V8 with *FseI* and *BsmI*. Fragment 2 contained the Bal31 deletion and was amplified from pRC.V8 using 5′-GCC AAG GGT TGG TTT GCG CAT TCA CAG TTC ACG ACG TTG TAA AAC GAC GGC CAG TGA AT-3′, containing a *BsmI* site, and 5′-CCG TTT CCT CCG GCG CCA TCT TCGC-3′. Fragment 3 contained the remaining pRC.V8 sequence and was amplified using 5′-GAG AAA CAA CTG TGC AGC GAG ATGC-3′ and 5′-CTT GCC CAG TCA GTC AGG GCC GGC CCA GGAG-3′. A non-coding mCherry sequence was then inserted into the *ClaI* site located after *env*.

### 2.2. XL-SHAPE, Primer Extension and Capillary Electrophoresis

RSV 5′-leader RNA (636 nt) was probed by SHAPE using N-methylisatoic anhydride (NMIA) as described previously with minor alterations [51]. Briefly, 2.3 µM RNA was folded in 50 mM HEPES, pH 7.5 by heating to 80 °C for 2 min, then 60 °C for 2 min. A final concentration of 1 mM MgCl_2_ was then added, followed by incubation at 37 °C for 15 min and incubation on ice for at least 30 min. The folded RNAs were diluted to 0.26 µM in 20 mM HEPES, pH 7.5, 1 mM MgCl_2_. Diluted RNA (9 µL) was mixed with either 1 µL of neat dimethyl sulfoxide (DMSO) or 1 µL of NMIA (80 mM in DMSO). The SHAPE reactions were incubated at 37 °C for 22 min and the RNAs were recovered by ethanol precipitation. 

Prior to XL-SHAPE experiments, the RNAs were folded as described above. The folded RNAs were diluted to 0.26 µM and incubated with Gag∆PR for 30 min at room temperature, in a buffer containing 20 mM HEPES, pH 7.5, 1 mM MgCl_2_, 10 mM KCl and 40 mM NaCl. Experiments were carried out with RSV Gag∆PR concentrations of 0.4, 0.27 and 0.18 µM in three separate reactions. For SHAPE probing, 1 µL NMIA (80 mM in DMSO) was added into 9 µL of each of the three binding reactions. Control reactions without RSV Gag∆PR contained either neat DMSO or NMIA (80 mM in DMSO). The SHAPE reactions were incubated at 37 °C for 22 min. The RNAs were recovered by phenol–chloroform extraction and ethanol precipitation. For XL experiments, each of the binding reactions (9 µL) was exposed to UV light (254 nm, total energy of 200 mJ/cm^2^) on ice in a Stratalinker 2400 UV cross-linker (Stratagene, San Diego, CA, USA). Two control XL experiments were also performed without RSV Gag∆PR: an RNA-only reaction was exposed to UV light to assess UV damage and a second RNA-only reaction was incubated on ice in the absence of UV irradiation. After XL, all the samples were treated with 1 µL of 5% SDS and 1 µL of Proteinase K (New England Biolabs, Ipswich, USA), and incubated at 55 °C for 60 min. The RNAs were recovered by phenol–chloroform extraction and ethanol precipitation. 

Following XL-SHAPE experiments, RNAs were resuspended in 9 µL of water and mixed with 1 µL of a mix of all four dNTPs (10 mM of each) and 2 µL of a 5′-6-FAM labeled primer (Applied Biosystems, Foster City, CA, USA). The primer annealing was achieved by incubating at 85 °C for 1 min, 60 °C for 5 min, 35 °C for 5 min. The reactions were then brought to 50 °C and 8 µL of reverse transcription mixture containing 1 µL of Superscript III (200 units/µL, Invitrogen), 4 µL of 5 × first-strand buffer, 2 µL of 0.1 M dithiothreitol (DTT) and 1 µL of water were added. Primer extension reactions were incubated at 55 °C for 1 h and then the enzyme was inactivated at 70 °C for 15 min. The RNAs were hydrolyzed using NaOH, as described [51]. Dideoxy sequencing reactions were performed on the template plasmid used for in vitro transcription, and all the reactions (XL-SHAPE, dideoxy sequencing) were analyzed by capillary electrophoresis, as described [51]. The sequences of the primers used in the primer extension reactions were as follows: 5′-AGC CGC CTT CAA TGC CC-3′ (Primer 1, complementary to nt 615–631 of RSV 5′-leader RNA), 5′-GGT TTT ACA CGC GGA CGA AAT CAC C-3′ (Primer 2, complementary to nt 397–421) and 5′-CCT CTA CTA GGG TCA TCG TCC GC-3′ (Primer 3, complementary to nt 190–212). At least three independent experiments were performed for each of the primers and each of the experimental conditions.

### 2.3. XL-SHAPE Data Analysis and RNA Structure Modeling

The raw data obtained from the capillary electrophoresis analysis were analyzed using the RiboCAT software, as described [51,52]. The individual XL and SHAPE reactivity data from each of the three primers were combined and averaged. The SHAPE data were used as pseudo-energy constraints for the structure modeling using RNAstructure [53]. A helix file was generated in RNAstructure and loaded into XRNA (http://rna.ucsc.edu/rnacenter/xrna/xrna.html, UCSC, USA), which was used to generate the secondary structure representation. The differences between the SHAPE reactivity of the RNA-only sample and the sample containing RSV Gag∆PR of the highest concentration were compared using an unpaired, two-tail student *t*-test. Absolute reactivity differences of ≥0.3 with a *p* value < 0.05 were considered to be statistically significant [54]. In the case of XL experiments, the differences between the reactivity of the RNA-only sample exposed to UV light and the sample containing RSV Gag∆PR of the highest concentration were analyzed in the same way as for the SHAPE assays. 

### 2.4. RNA Dimerization Assays 

The folding of WT and mutant RSV 5′-leader RNA for dimerization assays was conducted as follows: RNAs (0.6 µM) were incubated in 50 mM sodium cacodylate, pH 7.5 at 90 °C for 2 min followed by incubation on ice for 2 min. The RNAs were then diluted to 0.5 µM by adding low salt (50 mM sodium cacodylate, pH 7.5, 40 mM KCl and 0.1 mM MgCl_2_) or high salt (50 mM sodium cacodylate, pH 7.5, 300 mM KCl, and 5 mM MgCl_2_) buffers and incubated at 37 °C for 30 min, then on ice for 5 min. The samples were then mixed with 6 × native gel loading dye and run on a 1.2% native agarose gel containing 0.1 mM MgCl_2_ and no EDTA. The gel was run at 4 °C in TBM buffer (45 mM Tris borate buffer with 0.1 mM MgCl_2_) at 120 V for ~2 h. A 100 bp DNA ladder (New England Biolabs) was used as a migration reference to differentiate monomer and dimer RNAs. At least 2 independent experiments were performed.

### 2.5. Fluorescence Anisotropy (FA)-Based Salt Titration Binding Assays

RSV 167 and MΨ WT and mutants were folded and the FA salt titration binding assays were performed as previously described [45,48]. The final condition for these assays was as follows: 1.5 nM RNA, 20 mM HEPES, pH 7.5, 1 mM MgCl_2_, 50–1000 mM NaCl and 300 nM RSV Gag∆PR protein. 

### 2.6. Virus Infectivity Assays

QT6 quail fibroblast cells were seeded in 100 mm dishes and then transfected with 10 µg of the provirus using the calcium phosphate method. At 16 h post-transfection (hpt), the medium was changed. For the spreading infection assays in Figure 3B, cell lysates were collected at 3 and 6 days post-transfection (dpt) in RIPA buffer (50 mM Tris-HCl pH 7.2, 150 mM NaCl, 1% Triton x-100, 0.01% DOC, 0.1% SDS). The lysates were subjected to immunoblotting (50 µg of total protein as determined via Bradford assay). For the infection assays in Figure 3C, the medium was collected at 2 dpt. Half of the medium (used for normalization) was filtered through a 0.22 µM filter and spun through a 25% sucrose cushion at 55,000 rpm at 4°C in a TLA 100.4 rotor (Beckman Coulter, Brea, CA, USA) for 1 h. The virions were resuspended in PBS and transferred to preweighed 1.5 mL tubes and normalized by volume. Each sample (15 µL) was run on a 10% SDS-PAGE gel and transferred to a PDVF membrane. Immunoblotting against RSV CA was performed with a mouse anti-CA antibody (generated by Dr. Neil Christensen, Penn State College of Medicine) and detected via chemiluminescence. The intensity of each CA band was measured using the volume tool in ImageLab V4.1 (Bio-Rad, Hercules, CA, USA). The amount of virions in the other half of the medium collected from transfected cells was normalized to the sample with the lowest intensity band and added to naïve QT6 cells to perform infection. At 3, 6, 14 and 17 days post-infection (dpi), cell lysates were collected as above and genomic DNA (gDNA) was extracted using a PureLink Genomic DNA Mini Kit (ThermoFisher Scientific, Waltham, MA, USA). The lysates were subjected to immunoblotting (50 µg of total protein) similar to the spreading infection assays. To examine whether the mutations were maintained in infected cells, the gDNA collected from the infected cells was PCR amplified with Phusion polymerase (NEB) using primers 5′-GAT CGT CGA CGC CAT TTT ACC ATT CAC CAC ATT GGT GTG-3′ and 5′-GAT CGG TAC CGA TAG CAG GAT GTG CCA ACG GTT TTA GGTG-3′ and subjected to sequencing using primer 5′-GAT CGG GCC CGC TTG ATC CAC CGG GCG ACC-3′. Two biological replicates were performed.

## 3. Results

### 3.1. SHAPE Probing and Secondary Structure Analysis of RSV 5′-Leader

The secondary structures of several structural motifs of RSV gRNA, such as MΨ and primer binding site (PBS), have been previously predicted computationally [6,7,8,55]. However, whether these structural features exist in the context of the full 5′-leader has not been probed experimentally. Here, we used SHAPE to determine the secondary structure of the 636-nt RSV 5′-leader RNA. SHAPE investigates the local flexibility of all four ribonucleotides, showing higher 2′-hydroxyl group reactivity with less constrained or unpaired residues and lower reactivity with more constrained or base-paired residues [56,57]. The RSV 5′-leader was predominately monomeric under the probing reaction conditions (data not shown). NMIA was used for the probing and RiboCAT [52] was used to analyze the SHAPE data. To ensure optimal resolution and data coverage, three primers that are complementary to different regions within the 5′-leader RNA were used. A complete reactivity profile using the averaged data from all three primers is shown in Appendix A. To confirm that the use of multiple primers led to unbiased normalization of SHAPE reactivity, we checked the data consistency in the overlapping regions covered by more than one primer and observed high data consistency (Appendix A). The number of nucleotides covered by more than one primer accounted for 21% of the total number of nucleotides probed. 

The normalized SHAPE reactivity data were applied as pseudo-energy constraints during the structure modeling in RNAstructure [53] and the predicted lowest energy secondary structure of the RSV 5′-leader RNA is shown in Figure 1. Most of the nucleotides that showed high or very high SHAPE reactivity (colored in orange and red, respectively) were observed in the single-stranded regions. Nucleotides that showed intermediate SHAPE reactivity (colored in blue) were generally in a single-stranded region or in base-paired regions adjacent to loops and bulges. While the majority of nt in base-paired regions displayed low reactivity, we also observed a few nucleotides in single-stranded regions that displayed low reactivity (e.g., L3 loop) (colored in black). The L3 loop was previously determined to be the dimerization site of the closely related alpharetrovirus ALV [40,41] and the low SHAPE reactivity in this region is consistent with transient kissing-loop interactions between two RNA monomers. This potential dimerization site of RSV will be further examined and discussed below. 

The previously computationally predicted structures of MΨ and PBS [6,7,8,55] are highly consistent with our experimentally determined secondary structure model of the RSV 5′-leader RNA. The structural features of stem loops SL-A, SL-B, SL-C and L3 are identical, although the nucleotides at the 3′-end of the previously predicted MΨ structure are part of a larger stem loop in our structure (Figure 1). Our new structural model also revealed additional structural features within the PBS domain, and a long-range base-pairing interaction near the *gag* start codon (nt 145–150 and nt 386–391, Figure 1). In the case of HIV-1 and other retroviral 5′-leader RNAs, the AUG start codon is involved in long-range interactions [51,58,59,60,61]. In contrast, nucleotides at and near the AUG start codon of RSV’s 5′-leader RNA showed high SHAPE reactivity, indicating the absence of such long-range pairing interactions. 

### 3.2. The Dimerization Site of RSV 5′-Leader RNA is Bipartite

The palindromic sequences in the L3 loop were previously reported to be the DIS of ALV [40,41]. However, an antisense oligonucleotide directed against the L3 loop of the RSV 5′-leader only partially prevented RNA dimerization [40]. We hypothesize that more than one DIS is present in the RSV 5′-leader RNA. Our experimentally determined secondary structure revealed additional palindromic loop sequences, which are possible DIS candidates. The previously predicted DIS in the L3 loop (DIS-1) corresponds to nt 262–268 (Figure 1). A second palindromic loop is present in SL-A (nt 181–187)—UUCGGG (DIS-2) (Figure 1). Even though wobble base-pairing has not been reported in retroviral DIS sequences [36], similar to DIS-1, the DIS-2 loop has an unpaired adenosine at the start of the loop. This unpaired adenosine within DIS-1 is critical for ALV dimerization [40]. Additionally, the nucleotides in the middle of DIS-2 showed relatively low SHAPE reactivity (Figure 1), consistent with potential intermolecular interactions. 

To test whether the DIS candidates play a role in RSV RNA dimerization, we individually replaced the seven nucleotides in DIS-1 and DIS-2 with a stable GAGA tetraloop in the context of the 636-nt RSV 5′-leader RNA (∆DIS-1 and ∆DIS-2, Figure 2A). As a negative control, we constructed a mutant wherein the loop region from nt 445 to nt 447 was replaced with a GAGA tetraloop (∆DIS-0, Figure 2A). Finally, to examine the combined effect of loop mutations, we prepared a DIS-1/DIS-2 double GAGA mutant (DIS-DM), and a DIS triple mutant (DIS-TM) containing GAGA substitutions in all three loops (Figure 2A). The WT and all mutant RNAs were folded in the presence of low or high salt buffers, as described in the Materials and Methods, and their dimerization capability was analyzed using native agarose gel electrophoresis. The WT RSV 5′-leader RNA was primarily a monomer under low salt conditions, whereas it was predominately dimeric when folded in the presence of high salt (Figure 2B). The diffuse migration pattern of the dimer band suggests that there may be more than one conformation. In the case of the single mutants (∆DIS-1 or ∆DIS-2), the dimeric form was dominant in high salt, although a significantly more monomeric RNA was observed relative to the WT (Figure 2B). Under low salt conditions, the monomer was the major form observed for all RNAs, with the ∆DIS-1 mutant forming almost no dimer. The negative control ∆DIS-0 mutant migrated very similarly to the WT under all conditions (Figure 2B). Interestingly, the RNAs with the combined mutations migrated almost exclusively as monomers under both low and high salt conditions (DIS-DM and DIS-TM, Figure 2B). We also observed some smeared slowly migrating bands for the WT and single mutants under high salt conditions, which are likely to be higher-order oligomers.

We also made RNA mutants with extended stem loop deletions. As shown in Figure 2A, the ∆L3 and ∆SL-A constructs have deletions from nt 175–192 and nt 256–280, respectively. To minimize structural changes due to the deletions, the deleted sequences were replaced by the stable UCCG tetraloop. The combined effect was also examined using a double mutant ∆L3+SL-A. ∆L3 RNA showed a similar migration pattern as ∆DIS-1, with mostly monomers observed under low salt conditions and mostly dimers observed under high salt (Figure 2C). ∆SL-A RNA also migrated primarily as a dimer under high salt, with a mixture of monomers and dimers observed under low salt conditions (Figure 2C). The ∆L3+SL-A RNA showed exclusive monomers under both conditions (Figure 2C) showing a stronger inhibitory effect of these extended deletions on RNA dimerization relative to the loop-only mutations. This study indicates that the disruption of either DIS-1 or DIS-2 only leads to the partial disruption of RSV 5′-leader RNA dimerization, while the combination of the mutations leads to an almost complete disruption of dimerization. 

### 3.3. DIS Mutations Do not Have a Negative Effect on Virus Infectivity in Culture

To determine whether the DIS mutations affected virus replication, the 5′ LTR of pAT.V8 (WT or DIS mutants) was cloned into pRC.V8 (Figure 3A). A negative control, RC.V8 Bal31 mCherry, which does not produce Gag [50], was used (Figure 3A). QT6 cells were transfected with WT or mutant proviruses and cell lysates were collected at 3 and 6 dpt (Figure 3B). Immunoblots were performed to detect the presence of Gag expression in cell lysates as an indication of integration to assess infectivity (Figure 3B,C). For the RC.pAT.V8 ∆DIS-2, RC.pAT.V8 ∆DIS-1 and RC.pAT.V8 DIS-DM mutations, Gag was produced in cell lysates at all time points (Figure 3B). Interestingly, there was consistently a greater level of Gag expression in RC.pAT.V8 ∆DIS-2 compared with RC.pAT.V8 WT. 

We also tested the constructs with the extended stem loop deletions for virus replication. As the method used to assess infectivity was a spreading assay, we sought to determine whether virus particles produced by transfected cells could establish infection in naïve cells. Virus particles were collected at 2 dpt and normalized before infecting naïve QT6 cells (Figure 3C; see Materials and Methods for details). Lysates were collected at 3, 6, 14 and 17 dpi. Immunoblots were performed using anti-RSV serum. Once again, Gag was expressed in cell lysates at all time points, indicating that infection was established using a cell-free virus. At 3 dpi, the level of the Gag protein expression in the lysate appeared to be similar for the WT and mutants. Beginning at 6 dpi, the RC.pAT.V8 ∆SL-A mutant appeared to express a higher level of Gag compared with the WT control and the other mutants. By extracting and sequencing the genomic DNAs, we confirmed that the DIS mutations were maintained during infection. 

### 3.4. Identification of RSV Gag∆PR-Induced 5′-Leader RNA Conformational Changes and RSV Gag∆PR-Binding Sites

A combined crosslinking (XL) and SHAPE analysis was applied to study the interactions between the 636-nt RSV 5′-leader RNA and RSV Gag∆PR. This approach was used to distinguish between RNA structural changes induced by protein binding and the direct protein interaction sites [42]. The domains of RSV Gag∆PR are shown in Figure 4A and the XL-SHAPE results are summarized in Figure 4B. Experiments were conducted using three protein–RNA molar ratios (0.7:1; 1:1; 1.5:1). To ensure that the identified sites reflect specific effects of RSV Gag∆PR binding, reactivity changes were considered significant only if dose-dependent effects were observed (Appendix A, Figure 4B). 

Consistent with the central role of MΨ in directing gRNA packaging [6,7] and the binding specificity of RSV Gag∆PR to MΨ [45], SHAPE reactivity changes and XL sites were most prevalent in or near the MΨ region (Figure 4B, Appendix A). Previous NMR studies of an RNA derived from the 82-nt µΨ region showed that the SL-C loop adopts a canonical UNCG-type tetraloop [33] with the G218 residue largely exposed. This residue was shown to be a direct interaction site of RSV NC [33]. In good agreement with this previous study, we identified G218 as a strong Gag∆PR XL site (Figure 4B). Several additional Gag∆PR interaction sites in MΨ were also identified: U169, A205, G247 and A262. In the XL experiments, a strong bias towards a particular nucleotide identity was not observed, but purine residues were preferred and no XL was observed to cytosine. Nucleotide positions with significant SHAPE reactivity changes were distributed throughout the RNA with the highest density in the MΨ region; both increased and decreased flexibility changes were observed (Figure 4B). The lack of clustered SHAPE reactivity changes suggested that the overall conformation of MΨ was not globally altered upon binding of RSV Gag∆PR. Several SHAPE changes and XL sites occurred towards the beginning of MΨ, coinciding with the previously identified μΨ region (nt 161–234) (Appendix A). 

RSV Gag∆PR can chaperone the annealing of tRNA^Trp^ to the PBS of RSV gRNA [16]. A Gag XL site was observed at U102, which is adjacent to the start of the PBS (nt 102–119). We also observed one nt in the PBS with significantly increased SHAPE reactivity upon Gag∆PR binding (Appendix A and Figure 4B), indicating that protein binding leads to increased nt flexibility.

### 3.5. Mutations of Identified Crosslinked Residues in RSV MΨ Showed Distinct Effects on Gag∆PR Binding Specificity

We have previously used fluorescence anisotropy (FA) salt titration binding assays to characterize the binding of HIV-1 and RSV Gag proteins to Ψ-containing and non-Ψ RNAs, as well as to mutant Ψ RNA constructs [45,62]. These assays measure the salt dependence of protein–RNA interactions and provide information on the strength of the specific, non-electrostatic binding component (K_d,1M_), and the effective charge (Z_eff_), or the number of Na^+^ ions displaced during binding [48]. Here, we used salt titration binding assays to assess RSV Gag∆PR binding to the WT and mutant MΨ RNA. Based on the identified XL sites, we made five MΨ mutant RNAs: G218C, ^168^AUC^170^ to AAA (MΨ-AAA), ^244^GGCG^247^ to AACA (MΨ-AACA), ^262^ACUGCAG^268^ to UCCG (MΨ-∆L3) and ^205^AGUAG^209^ to UCCG (MΨ-∆SL-B) (Figure 5A). Additionally, we chose to test an A197C mutant because it was shown to be an NC interaction site in an NMR study [33]. These RNAs ranged in length from 164 to 167 nt and a 167-nt RNA, RSV 167, derived from the *gag* gene was used as a non-specific RNA control, as previously described [45]. 

In agreement with our previous work [45], RSV Gag∆PR binding to MΨ-WT was characterized by stronger non-electrostatic interactions and fewer electrostatic contacts (K_d,1M_ = 2.3 × 10^−4^ M and Z_eff_ = 4.5) than binding to RSV 167 (K_d,1M_ = 2.1 M and Z_eff_ = 10.1) (Figure 5B,C, Table 1). Among the MΨ mutants that were tested, MΨ-G218C led to the largest decrease in Gag∆PR binding specificity, with a K_d,1M_ of 7.0 × 10^−1^ M and Z_eff_ of 9.1 (Figure 5B,C, Table 1). This led to a reduction in binding specificity to a level that is similar to the non-specific RSV 167 RNA (Table 1). RSV Gag∆PR binding to MΨ-A197C, MΨ-AAA and MΨ-∆L3 exhibited a significant, but modest reduction in binding specificity (K_d,1M_ ranged from 3.5 × 10^−3^ to 3.1 × 10^−2^ M and Z_eff_ values were ~6–7) (Figure 5B,C, Table 1). These corresponded to 15~135-fold differences in the relative binding specificity for these mutants (Table 1). Finally, Gag∆PR binding to MΨ-AACA and MΨ-∆SL-B was not significantly different from that of WT MΨ (Figure 5B,C, Table 1).

## 4. Discussion 

The 5′-leader region of RSV gRNA regulates numerous aspects of the life cycle including gRNA packaging, which is highly dependent on the µΨ region embedded within the larger MΨ sequence [6,7,8]. To date, only phylogenetic and computational approaches have been used to predict a consensus secondary structure model of this region of the genome [63]. Here, we probed the secondary structure of the RSV 5′-leader RNA experimentally for the first time using SHAPE. Our SHAPE-derived secondary structure suggests that RSV µΨ is similar to the predicted consensus model in the context of the longer construct (Figure 1). Our structure of MΨ is very similar to the computationally predicted model, except for the 3′-end (nt 307–316) [6,7,8]. The major splice donor (nt 397) of RSV gRNA is located downstream of MΨ and consequently, RSV spliced viral RNAs contain MΨ. However, the MΨ-containing spliced viral *env* mRNA is not efficiently packaged [7]. A possible explanation is that the MΨ structure in the *env* mRNA differs from that of unspliced gRNAs or efficiently packaged MΨ-containing heterologous RNAs. This may be the mechanism by which spliced viral RNAs are excluded from packaging. Solving the secondary structure of the 5′-leader region of spliced *env* mRNA in the future would test this hypothesis. We also observed several stem loops downstream of the *gag* start codon and nt 488–632 are predicted to form a long and structured region. Whether these structural motifs have important biological functions remains to be elucidated.

Another key step that the RSV 5′-leader RNA regulates is gRNA dimerization. We identified a bipartite DIS in the 5′-leader and disruption of either DIS does not lead to the complete disruption of RNA dimerization (Figure 2B,C). We did not observe a large proportion of higher-order RNA oligomers, suggesting that these two DISs are functionally redundant. A bipartite dimerization signal has been identified previously in human T-cell leukemia virus type 1 (HTLV-1) [51] and Moloney murine leukemia virus (MMLV), another simple retrovirus [64]. However, some other retroviruses, such as HIV-1, only have one DIS [36]. Therefore, different retroviruses have evolved to adopt different mechanisms to achieve gRNA dimerization [36]. In ALV, the DIS in the L3 loop, which corresponds to DIS-1 in RSV, is of primary importance in initiating dimerization [40,41]. Since µΨ contains DIS-2 but not DIS-1, if dimerization is a prerequisite for the efficient packaging of heterologous RNAs containing RSV µΨ [8], then DIS-2 may be sufficient. However, an early study showed that RSV *env* mRNA still dimerized efficiently in vitro even though it is not abundantly packaged [40]. 

In the context of a full-length provirus, these DIS mutations do not impair RSV virus replication (Figure 3). These data suggest either that these regions are not necessary for dimerization in cell culture, or that sequences outside of this region contribute to dimerization and packaging in the context of a proviral vector. Efficient RNA dimerization was also not strictly required for HTLV-1 gRNA packaging [51], and HIV-1 can also replicate without the DIS [65,66]. Therefore, retroviral gRNA dimerization propensity in vitro does not appear to be a good predictor of retroviral packaging selectivity in cells or viral infectivity. Interestingly, both the RC.pAT.V8 ∆DIS-2 and RC.AT.V8 ∆SL-A mutants, which expressed higher levels of Gag, contain mutations in the same region (Figure 2A), suggesting that the deletion of these sequences could enhance Gag production by altering the RNA transcription, stability, translation efficiency through the secondary structure or long-range interactions of viral RNA.

Previous studies of retroviral 5′-leader sequences in HIV-1 [58,59], HTLV-1 [51], Mason-Pfizer monkey virus (MPMV) [60], mouse mammary tumor virus (MMTV) [61] and feline immunodeficiency virus (FIV) [67] have predicted long-range interactions involving the AUG start codon. In HIV-1, this long-range interaction is proposed to regulate the conformation adopted by the 5′-UTR and determine the fate of gRNA for either translation or packaging [59]. Based on the high SHAPE reactivity of nucleotides at and near the AUG start codon, our data do not support the existence of a long-range interaction. This difference between RSV and HIV-1 may be due to distinct mechanisms of gRNA packaging. While initial HIV-1 gRNA recognition likely occurs in the cytoplasm [68,69], initial RSV gRNA recognition likely occurs in the nucleus via nuclear trafficking of Gag [70,71,72]. 

Based on our SHAPE-derived structure of the RSV 5′-leader RNA, the PBS region is highly structured (Figure 1). Interestingly, even the 7-nt single-stranded bulge within the PBS showed very low SHAPE reactivity, indicating the possibility of long-range tertiary interactions. In addition, some nucleotides in the double-stranded region near the PBS showed high SHAPE reactivity, indicating this region is dynamic. It is likely that tRNA primer annealing stabilizes the overall conformation of the PBS. Indeed, previous studies showed that the PBS undergoes significant structural changes and forms extensive intermolecular interactions upon primer annealing [55]. The tRNA^Trp^ primer is found annealed to the PBS in protease-defective virions, indicating that the Gag precursor is capable of chaperoning the annealing [21]. Previously, we reported that RSV Gag is a very efficient chaperone on a molar basis [16]. The strong chaperone activity of RSV Gag may be beneficial in facilitating tRNA^Trp^ annealing to the highly structured PBS. 

The initiation of RSV gRNA packaging relies on Gag–Ψ interactions. To our knowledge, the XL-SHAPE studies reported here are the first comprehensive analysis of interactions between RSV Gag and the RSV 5′-leader RNA (Figure 4). In this analysis, the Gag concentration was kept low relative to RNA, in order to determine the most specific binding effects. Consistent with the central role of MΨ in RSV gRNA packaging, most of the identified binding sites or SHAPE reactivity changes are in the MΨ region. A previous study indicated that μΨ adopts more than one conformation and NC binding possibly stabilizes one of the conformations [33]. In agreement with this idea, we observed SHAPE reactivity changes in the μΨ region upon Gag binding (Figure 4). 

Based on the FA salt titration binding assays, we identified G218 in SL-C as critical for Gag binding specificity (Figure 5). This is in good agreement with NMR studies that identified G218 as a direct NC interaction site [33]. The single point mutation tested here results in a UGCG to UCCG loop change. Although a GAGA mutation in this loop, which changes G218 to A, was previously found to partially reduce viral infectivity [33], this mutation does not affect Gag binding specificity in vitro [45]. Thus, a purine to pyrimidine change at 218 appears to be more detrimental to Gag recognition in vitro than a G to A change. Another site identified as an NC interaction site in the NMR studies [33], A197, was not identified in our XL experiments. According to the NMR structure, an NC-A197 interaction involves multiple hydrogen bonds and salt bridges [33]. It is possible that the A197 interaction was not efficiently crosslinked due to the amino acid dependence of the UV XL method [73]. The A197C mutation did significantly impact Gag binding specificity in our FA binding assays, although not as strongly as the G218C mutation.

Another Gag interaction site that was identified in the XL experiments, U169, was tested by making the MΨ-AAA variant, which affected Gag binding specificity moderately (Figure 5). This is consistent with assays showing that the substitution of AUC with CCU abolished viral infectivity [33]. We also showed that the replacement of the L3 loop by a tetraloop slightly decreased Gag binding specificity (Figure 5). However, this site may not be critical since the L3 stem loop is dispensable for packaging [8]. Among all the interaction sites that contribute to Gag binding specificity, three out of four are located in µΨ. Previously, we identified three G-rich single-stranded sites in HIV-1 Ψ that are important for high HIV-1 Gag binding specificity and these three sites cluster together according to a SAXS model [45,74]. No single mutation can reduce the Gag binding specificity to the level of non-Ψ RNAs and we proposed that multiple interaction sites in close proximity are critical for specific binding [45]. It is likely that these binding sites also cluster together in µΨ, which could promote Gag oligomerization and viral assembly. Future studies using RNA constructs containing combined mutations, as well as additional structural investigations, are needed to test these ideas.

In summary, we obtained the SHAPE-derived secondary structure of a 636-nt RSV 5′-leader RNA for the first time. Using this structure, we predicted and identified a bipartite DIS in the MΨ region. Although mutations of both DIS loops abolished dimerization in vitro, these same changes failed to reduce viral replication, indicating that mutations that impair dimer formation using in vitro methods must be tested in full length proviral constructs to establish their functional significance in cells. Using high-throughput XL-SHAPE, we also identified numerous Gag interaction sites, some of which are novel. G218 contributes the most to Gag binding specificity, which is in good agreement with previous studies performed with RSV NC [33]. Although the SHAPE analysis performed here probes the secondary structure, it is likely that the RNA tertiary structure also plays a role in Gag interaction and packaging, as suggested by the NC–µΨ complex probed by NMR [33]. Overall, our results provide new insights into the RSV gRNA secondary structure, dimerization and Gag interactions with important implications for retroviral gRNA packaging.

## Figures and Tables

**Figure 1 viruses-12-00568-f001:**
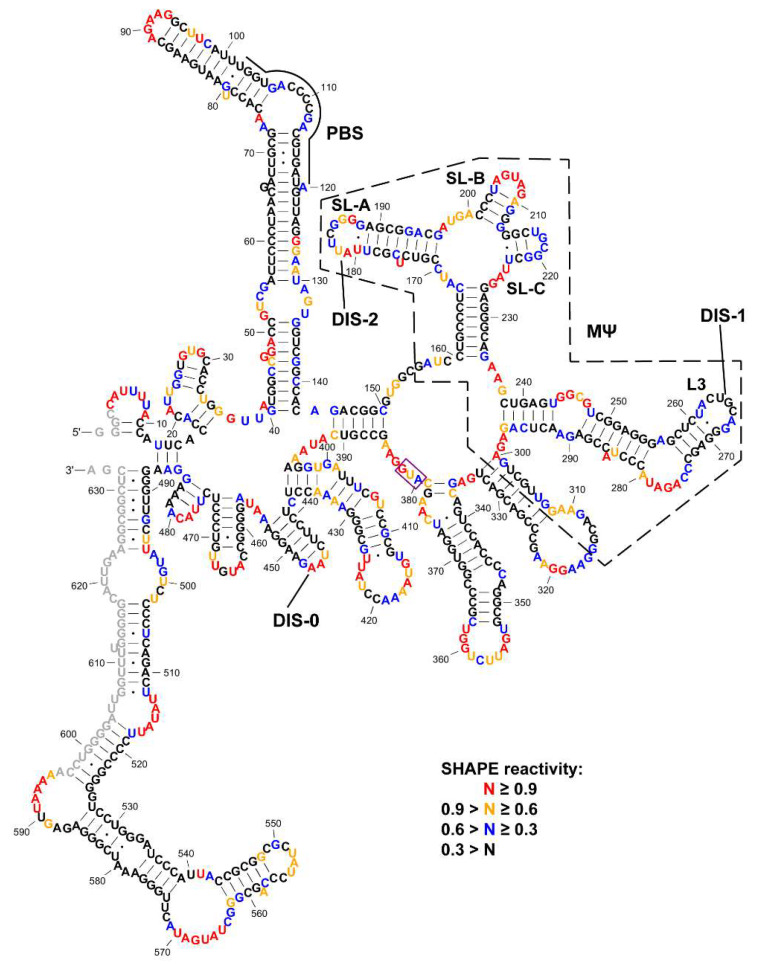
Selective 2′-hydroxyl acylation analyzed by primer extension (SHAPE) reactivity-constrained lowest energy secondary structure of the 636-nt Rous sarcoma virus (RSV) 5′-leader RNA. The secondary structure model was generated by applying averaged normalized SHAPE reactivity from three independent trials as pseudo free-energy constraints in RNAstructure [53]. Nucleotides are colored in accordance to SHAPE reactivity as indicated in the key. Nucleotides that could not be analyzed due to primer annealing (3′ end) or the lack of resolution (5′ end) are shown in grey. MΨ is indicated by a black dashed box. Stem loops in µΨ and loops that were tested in the dimerization assays (DIS-0, DIS-1/L3, DIS-2/SL-A, SL-B and SL-C) are labeled. The AUG start codon is indicated by a purple box.

**Figure 2 viruses-12-00568-f002:**
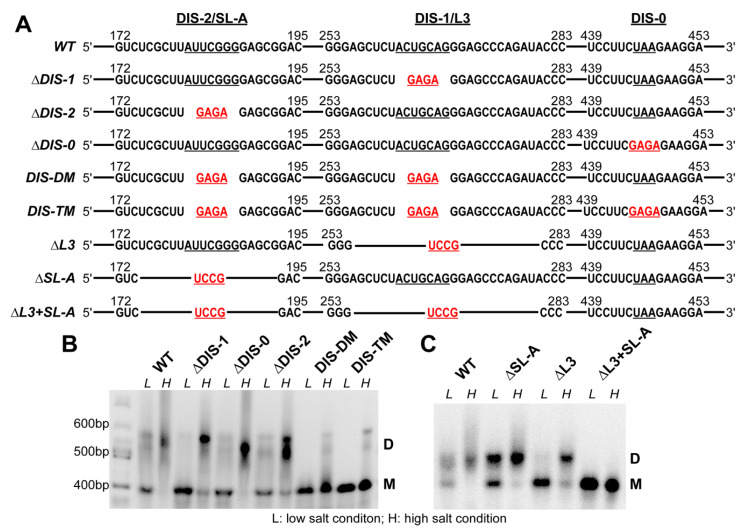
(**A**) RNA constructs used in the dimerization studies. Sequences in the regions containing DIS-2/SL-A, DIS-1/L3 and DIS-0 are shown. Nucleotides in the loop regions are underlined. Tetraloop mutations are highlighted in red. (**B**) Native agarose gel showing the results of dimerization assays of the wild-type (WT) RSV 5′-leader and dimerization initiation signal (DIS) loop mutants. RNAs (0.5 μM) were folded under low salt (L) or under high salt (H), as described in the Materials and Methods. Locations for monomeric (M) and dimeric RNA bands (D) are indicated. This gel shown is representative of three independent replicates. (**C**) Native agarose gel showing the results of dimerization assays of the WT RSV 5′-leader and more extensive stem loop deletion mutants. The conditions are the same as panel (B). This gel shown is representative of two independent replicates.

**Figure 3 viruses-12-00568-f003:**
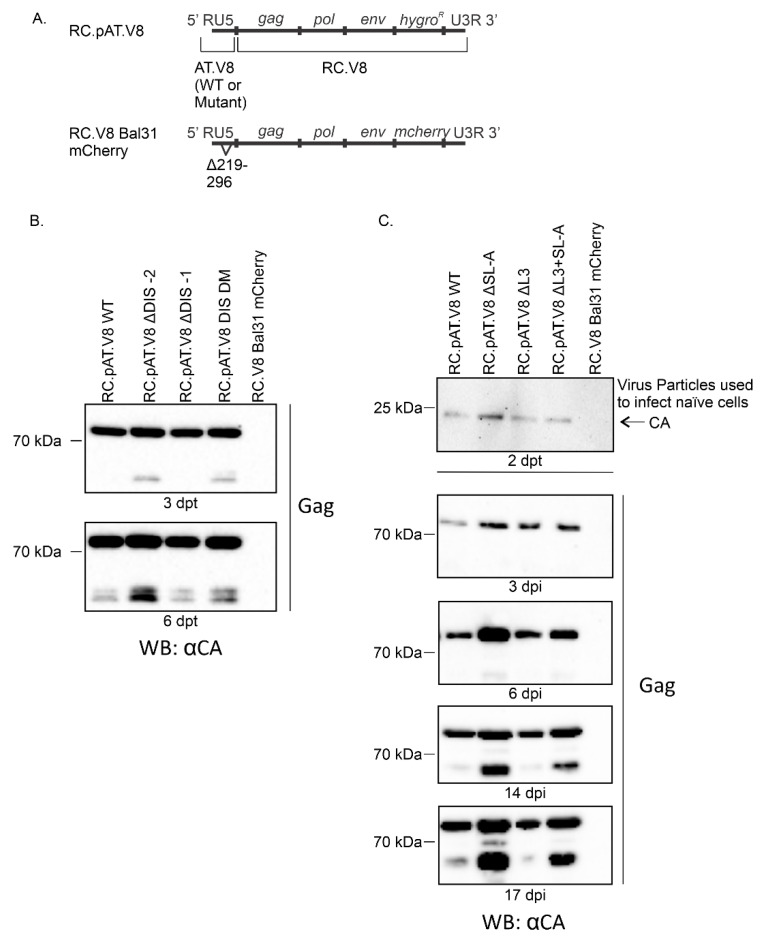
(**A**) Schematic of the constructs used in the infectivity studies. (**B**) Immunoblots of Gag in 50 µg of lysates collected from QT6 cells transfected with RC.pAT.V8 WT, RC.pAT.V8 ∆DIS-2, RC.pAT.V8 ∆DIS-1, RC.pAT.V8 DIS-DM or RC.V8 Bal31 mCherry. Lysates were collected in RIPA at 3 and 6 dpt. (**C**) Immunoblots of virions collected from QT6 cells transfected with RC.pAT.V8 WT, RC.pAT.V8 ∆SL-A, RC.pAT.V8 ∆L3, RC.pAT.V8 ∆L3+SL-A and RC.V8 Bal31 mCherry (top panel, 2 dpt), and immunoblots of Gag in 50 µg of lysates collected from QT6 cells infected with viruses produced from the transfected cells (bottom panel, 3, 6, 14 and 17 dpi).

**Figure 4 viruses-12-00568-f004:**
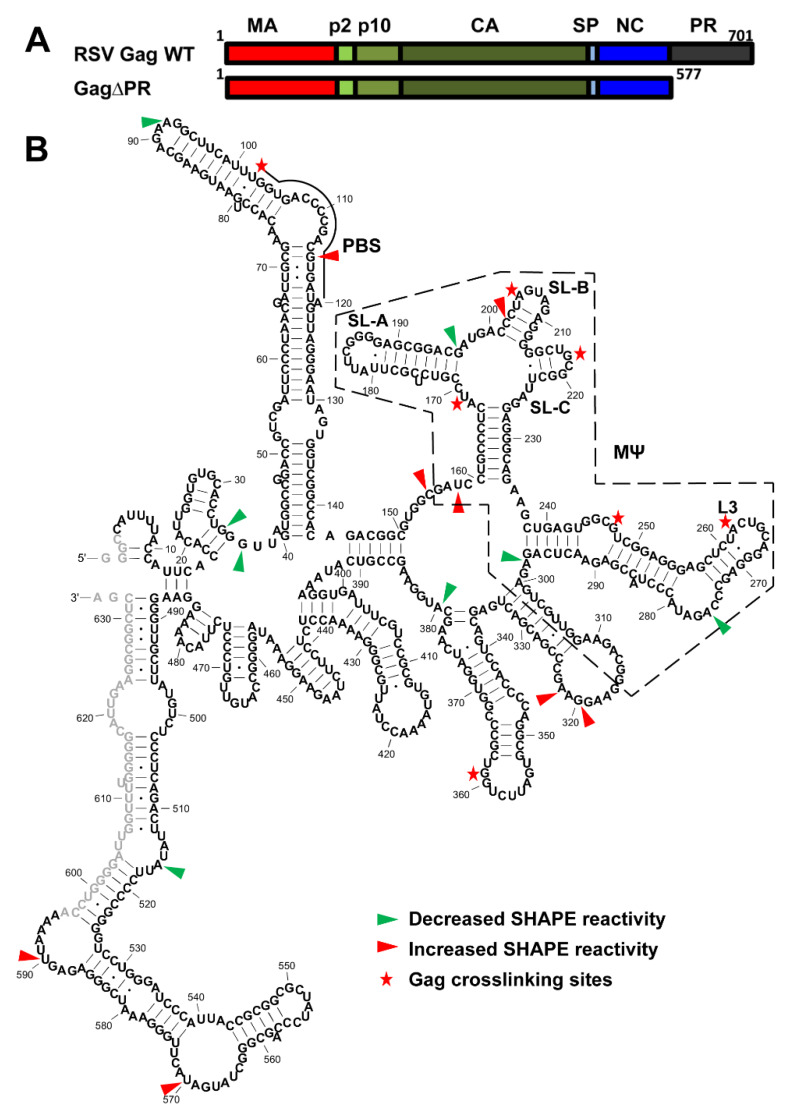
Crosslinking (XL)-SHAPE analysis of RSV Gag∆PR binding to the RSV 5′-leader RNA. (**A**) Schematic showing individual domains of RSV Gag and RSV Gag∆PR. (**B**) XL-SHAPE results mapped to the secondary structure of the RSV 5′-leader RNA. Sites with decreased and increased SHAPE reactivity upon protein binding are indicated by green and red arrows, respectively. Identified crosslinking sites are labeled with stars. All identified sites have reactivity changes of ≥0.3 and *p* < 0.05 based on unpaired, two-tailed student *t*-tests, compared with the no protein control. The MΨ region (dashed box) and PBS are indicated. Results are based on the average of at least 3 independent experiments.

**Figure 5 viruses-12-00568-f005:**
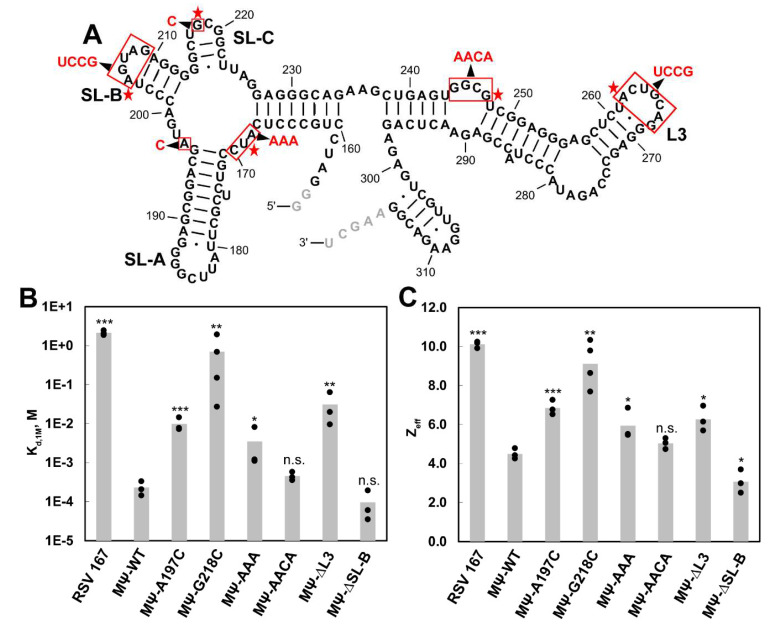
(**A**) Secondary structure of RSV MΨ. Six mutant MΨ constructs tested are indicated by red boxes and black arrows. Additional nucleotides that were added to facilitate in vitro transcription are shown in grey. (**B,C**) Bar graphs of K_d,1M_ (B) and Z_eff_ (C) values determined by fluorescence anisotropy (FA) salt titration binding assays using RSV Gag∆PR and RSV 167, MΨ-WT and six MΨ mutants. Values of three to four trials are indicated by black dots and the average is indicated by the height of the bar. *p* values were determined in comparison with MΨ-WT using unpaired, two-tailed student *t*-tests. (***) *p* < 0.001; (**) *p* < 0.01; (*) *p* < 0.05; (n.s.) not significant.

**Table 1 viruses-12-00568-t001:** K_d,1M_ and Z_eff_ values determined by FA salt titration binding assays of RSV Gag∆PR with RSV 167, MΨ-WT and six MΨ mutants.

RNA	K_d,1M_ ^1^	Z_eff_ ^2^	Relative Specificity ^3^
RSV 167	(2.1 ± 0.3) × 10^0^	10.1 ± 0.2	0.00011
MΨ-WT	(2.3 ± 1.0) × 10^−4^	4.5 ± 0.3	1.0
MΨ-A197C	(9.8 ± 4.0) × 10^−3^	6.8 ± 0.4	0.023
MΨ-G218C	(7.0 ± 8.8) × 10^−1^	9.1 ± 1.2	0.00033
MΨ-AAA	(3.5 ± 4.1) × 10^−3^	5.9 ± 0.8	0.066
MΨ-AACA	(4.5 ± 1.1) × 10^−4^	5.0 ± 0.3	0.51
MΨ-∆L3	(3.1 ± 2.9) × 10^−2^	6.3 ± 0.6	0.0074
MΨ-∆SL-B	(9.6 ± 8.5) × 10^−5^	3.1 ± 0.6	2.4

^1^ K_d,1M_ reflects the specific, non-electrostatic binding component. ^2^ Z_eff_ represents the effective charge of a protein–RNA interaction. ^3^ Relative binding specificity of the RNAs was calculated by K_d,1M(MΨ-WT)/_ K_d,1M(mutant)_ with the MΨ-WT value set to 1.0.

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
