# Peer review of "Rous Sarcoma Virus Genomic RNA Dimerization Capability In Vitro Is Not a Prerequisite for Viral Infectivity"

_viruses, 2020, doi:10.3390/v12050568_

Round 1
Reviewer 1 Report
This paper represents a significant advance in our understanding of RSV gRNA dimerization and function. The work is carefully and thoroughly performed and the emerging experimentally validated structural information on the RSV leader sequence is particularly valuable.
In order to make the data more accessible to a broader readership, I recommend that the authors provide the following additional information:
- “Heterologous RNA” as used in the beginning of the paper needs to be defined. The presumption is that this refers to gRNA that is not contained in the infecting virion. But “heterologous RNA” could refer to many other possible gRNAs, including those from any other retrovirus that has infected the same cell.
- A more detailed, accurate and complete description of vectors used in bacterial as well as animal cells is essential with appropriate references, not some historical papers that are about 40 years old but with recent and well-described use of these vectors. Materials and Methods must provide the necessary information for reproduction of the work described, and in this respect the current paper is defective.
- The techniques for measuring infectivity are indirect. If they are to provide more than a qualitative yes or no answer, statistical data on their accuracy need to be provided.
Author Response
- “Heterologous RNA” as used in the beginning of the paper needs to be defined. The presumption is that this refers to gRNA that is not contained in the infecting virion. But “heterologous RNA” could refer to many other possible gRNAs, including those from any other retrovirus that has infected the same cell.
Response: The concept of “heterologous RNA” was used in Banks et al., J. Virol. 1999 (ref 7) wherein a non-RSV RNA containing the Mpsi sequence was packaged into avian retroviral particles. At the first mention of “heterologous RNA” in the text, we now included (i.e., non-native).
- A more detailed, accurate and complete description of vectors used in bacterial as well as animal cells is essential with appropriate references, not some historical papers that are about 40 years old but with recent and well-described use of these vectors. Materials and Methods must provide the necessary information for reproduction of the work described, and in this respect the current paper is defective.
Response: A more recent reference was added describing the use of the RSV Gag expression construct, pET28(-His).RSVGag∆PR in line 108, and more detail for the cloning of pRC.V8 Bal31 mCherry was added (lines 151-152)
- The techniques for measuring infectivity are indirect. If they are to provide more than a qualitative yes or no answer, statistical data on their accuracy need to be provided.
Response: The method used to detect infectivity is not quantitative but gives a qualitative yes or no answer.
Reviewer 2 Report
The manuscript by Liu et al, studied the structure of RSV packaging signal by SHAPE method. Even though RSV has been studied for more than a century now, the molecular mechanism of its packaging is still not understood.
This article shows incremental improvement in understanding by identifying two putative dimerization initiation signals (DIS). However, the disruptions of both DIS was required to reduce dimerization in vitro. But the in vivo effects were not promising which diminishes the importance of the study. Mutations in DIS failed to affect replication in vivo.
The specific comments are-
I do not think the second part of last sentence in Abstract (specific protein-RNA interactions……. Therapy) should be there. This manuscript is way behind the identification of such sites and seem to be over-selling the conclusions from this study.
Figure 3; western blots is the size marker for second panel on Fig 3C correct. All other panels are gag band higher than 70 kd. What is second band of smaller size seen in blots at 14 and 17dpi as well as in Fig. 3A.
Author Response
- I do not think the second part of last sentence in Abstract (specific protein-RNA interactions……. Therapy) should be there. This manuscript is way behind the identification of such sites and seem to be over-selling the conclusions from this study.
Response: We have changed the last sentence to “Overall, our results suggest that disruption of either of the DIS sequences does not reduce virus replication and reveal specific sites of Gag-RNA interaction.
- Figure 3; western blots is the size marker for second panel on Fig 3C correct. All other panels are gag band higher than 70 kd. What is second band of smaller size seen in blots at 14 and 17dpi as well as in Fig. 3A.
Response: The size marker was corrected in figure 3C. The smaller band is a proteolytic cleavage product of Gag that contains CA.
Reviewer 3 Report
Liu et al.
The authors use biophysical and biological approaches to further understand the mechanism of retroviral gRNA packaging. It is generally believed is that retroviral gRNA is packaged through interactions between the Gag polyprotein and a psi packaging signal located in the 5'-UTR. The authors are studying RSV as a prototype packaging system. The psi RNA is predicted to be highly structured, with stem-loops contributing to gRNA dimerization and interaction with the viral Gag protein. Using an acylation-primer extension method the authors probed the secondary structure of the RSV 5'-leader RNA “for the first time”. They also identify a bipartite dimerization initiation signal (DIS) in vitro, and mutation of both sites was required to significantly reduce dimerization in vitro. However, these sequences were not required for viral replication. A UV crosslinking approach was also used to examine Gag protein-induced RNA conformational changes. G218 was identified as a critical Gag-interaction residue. Overall, the results suggest that disruption of either of the DIS sequences does not reduce virus replication and that specific protein-RNA interaction sites may be useful targets for viral inhibition. In summary, the manuscript investigates several distinct and key features of the RSV packaging mechanism.
Specific comments:
- line 26 “suggesting that in vitro dimerization results do not strictly correlate with in vivo infectivity”. This concept needs to be clarified throughout the manuscript. Do the results mean that gRNA dimerization per se is not required for replication? What is the expectation regarding the packaged RNA carrying the mutations? Dimer?
- The DLS and DIS should be more clearly defined, historically. Are they conceptually different?
- To strengthen the manuscript, it might be helpful to emphasize the derivation of the structural models. The authors describe an experimentally informed model. It is confusing to state in the Abstract that the leader structure has been determined “for the first time”. Yet, the authors refer to various loops (by modeling only) as if they were already well-defined. Also, line 392 describes key findings in the literature using NMR. It might be helpful to discuss carefully the number of lines of evidence that contribute to the various models. Otherwise, the methods described here sound confirmatory.
- line 45. “initiated by the interaction between the nucleocapsid (NC) domain of Gag and a packaging signal (psi), which is located in the 5'-untranslated region (UTR) of the gRNA [2-4].”
Are references 2-4 correct? I believe that a paper by Katz et al JVI 1986 first mapped the RSV packaging sequence.
- Does the method described here detect tertiary RNA structure? Are there reasons to include or exclude a major role for tertiary structure in packaging?
Author Response
- line 26 “suggesting that in vitro dimerization results do not strictly correlate with in vivo infectivity”. This concept needs to be clarified throughout the manuscript. Do the results mean that gRNA dimerization per se is not required for replication? What is the expectation regarding the packaged RNA carrying the mutations? Dimer?
Response: The results do not suggest that dimerization is not required for infectivity. The deletions of the sequences that impair dimerization in vitro do not have the same effect in vivo possibly due to long range interactions in the RNA. The expectation regarding the packaged RNA in infectious viruses would be dimers.
To address this comment, we changed the sentence in the abstract (line 26-28) to: “These mutations failed to reduce viral replication, suggesting that in vitro dimerization results do not strictly correlate with in vivo infectivity, possibly due to additional RNA interactions that maintain the dimers in cells.”
- The DLS and DIS should be more clearly defined, historically. Are they conceptually different?
Response: They are conceptually similar where DIS is a part of the DLS. In lines 88-89 we now write: “Previous in vitro dimerization studies in RSV characterized a dimer linkage structure (DLS), which was historically the name used to represent regions involved in gRNA dimerization within the gag coding region [37-39].”
- To strengthen the manuscript, it might be helpful to emphasize the derivation of the structural models. The authors describe an experimentally informed model. It is confusing to state in the Abstract that the leader structure has been determined “for the first time”. Yet, the authors refer to various loops (by modeling only) as if they were already well-defined. Also, line 392 describes key findings in the literature using NMR. It might be helpful to discuss carefully the number of lines of evidence that contribute to the various models. Otherwise, the methods described here sound confirmatory.
Response:
To address this comment, we modified the Abstract to read: “….we probed the secondary structure of the entire RSV 5'-leader RNA for the first time.”
At the beginning of Results, we now state (added the word “full” for emphasis): “The secondary structures of several structural motifs of RSV gRNA, such as MΨ and primer binding site (PBS), have been previously predicted computationally [6-8,55]. However, whether these structural features exist in the context of the full 5'-leader has not been probed experimentally.”
The NMR studies are also of a much shorter construct. The length of the construct studied by NMR is now given on line 394: “Previous NMR studies of an RNA derived from the 82-nt µΨ region…”
- line 45. “initiated by the interaction between the nucleocapsid (NC) domain of Gag and a packaging signal (psi), which is located in the 5'-untranslated region (UTR) of the gRNA [2-4].” Are references 2-4 correct? I believe that a paper by Katz et al JVI 1986 first mapped the RSV packaging sequence.
Thank you for bringing this important point to our attention. We have corrected the references and included two pioneering papers: Katz et al JVI 1986 and Meric et al JVI 1986.
- Does the method described here detect tertiary RNA structure? Are there reasons to include or exclude a major role for tertiary structure in packaging?
This is a good point and yes, tertiary structure may also play a role and is not addressed directly by our approach. We have now added a sentence starting on line 558 in the final paragraph: “Although the SHAPE analysis performed here probes secondary structure, it is likely that RNA tertiary structure also plays a role in Gag interaction and packaging, as suggested by the NC-µΨ complex probed by NMR [33].”